# Two Onnamide Analogs from the Marine Sponge *Theonella conica*: Evaluation of Geometric Effects in the Polyene Systems on Biological Activity

**DOI:** 10.3390/molecules28062524

**Published:** 2023-03-09

**Authors:** Fumiaki Nakamura, Hiroshi Kimura, Nobuhiro Fusetani, Yoichi Nakao

**Affiliations:** 1Department of Chemistry and Biochemistry, Graduate School of Advanced Science and Engineering, Waseda University, 3-4-1 Okubo, Shinjuku-ku, Tokyo 169-8555, Japan; 2Cell Biology Center, Institute of Innovative Research, Tokyo Institute of Technology, 4259 Nagatsuta, Midori-ku, Yokohama 226-8501, Japan; 3Research Institute for Science and Engineering, Waseda University, 3-4-1 Okubo, Shinjuku-ku, Tokyo 169-8555, Japan

**Keywords:** onnamides, *Theonella* sp., marine sponges, histone modifications, cytotoxicity, marine natural products

## Abstract

Two previously unreported onnamide analogs, 2*Z*- and 6*Z*-onnamides A (**1** and **2**), were isolated from the marine sponge *Theonella conica* collected at Amami-Oshima Is., Kagoshima Prefecture, Japan. Structures of compounds **1** and **2** were elucidated by spectral analysis. Structure–activity relationships (SARs) for effects on histone modifications and cytotoxicity against HeLa and P388 cells were characterized. The geometry in the polyene systems of onnamides affected the histone modification levels and cytotoxicity.

## 1. Introduction

Marine invertebrates are a rich source of compounds with unique structures and biological activities. Marine invertebrates are composed of the phyla Porifera, Cnidaria, Mollusca, Echinodermata, Chordata and so forth. Marine sponges of the genus *Theonella* are well known for their rich secondary metabolites, including nonribosomal peptides, polyketides and terpenoids [1,2,3,4,5,6]. Onnamides or theopederins share a common core skeleton produced via polyketide and nonribosomal peptide biosynthetic pathways [7,8,9], and form a group with distinct structures and potent cytotoxicity against cancer cell lines. In this study, the isolation and structure elucidation of two unreported onnamide analogs, 2*Z*- and 6*Z*-onnamides A (**1** and **2**), are described, as well as known analogs **3**–**6**.

Histone modifications play a crucial role in the epigenetic control of gene expression [10,11,12], and perturbations in this gene-switching system are related to chronic diseases such as cancer [13]. From this viewpoint, we have developed an in vitro cell-based assay system for evaluating the effects of compounds on multiple histone modifications in parallel [14]. Among 3750 extracts of marine organisms tested [14], the hydrophobic extract prepared from a marine sponge *T. conica* collected at Amami-Oshima Is., Kagoshima Prefecture, Japan, markedly enhanced the levels of trimethylated histone H3 lysine 27 (H3K27me3) and reduced the level of acetylated H4 lysine 5 (H4K5ac). Bioassay-guided isolation allowed us to identify onnamide analogs as active components in this marine sponge. Structure–activity relationships (SARs) for the activity of controlling histone modifications as well as for cytotoxicity were examined using six analogs (**1**–**6**). As the result, we observed that some of the histone modification changes occurred at the lower concentration than IC_50_ values for cytotoxicity. This finding imply different modes of action for these two biological activities that, however, were not fully supported because of the different sensitivity of HeLa cells to the compounds in the respective assays.

Therefore, our conclusion was that the activity in control of histone modification and cytotoxicity changes depending on the geometric isomerism in the side chains, but the different modes of action between these biological activities are not confirmed. Identification of the target gene expressions controlled by the histone modifications affected by onnamides should provide new insights into the underlying mechanisms of action for onnamides. A detailed investigation is currently underway.

## 2. Results and Discussion

A frozen specimen of *T. conica* (1020 g wet wt.) collected at Amami-Oshima Is., Kagoshima pref., Japan was extracted with CH_3_OH. The combined CH_3_OH extract was evaporated in vacuo and subjected to solvent partitioning [15], followed by ODS and SiO_2_ column chromatography. The obtained cytotoxic fractions were separated by repetitive reversed-phase HPLC on an ODS column to yield a mixture of 2*Z*- and 6*Z*-onnamides A (**1**, **2**), onnamide A (**3**) [2], 4*Z*-onnamide A (**4**) [4], dihydroonnamide A (**5**) [3] and onnamide B (**6**) [3]. Final purification of the mixture of **1** and **2** was performed by recycling reversed-phase HPLC on a PHENYL-HEXYL column to afford **1** (3.5 mg) and **2** (1.5 mg) (3.4 × 10^−3^% and 1.5 × 10^−3^% yield based on wet weights). All structures of **1**–**6** were determined by ^1^H NMR and MS analyses (Appendix A, Appendix A).

The molecular formula of 2*Z*-onnamide A (**1**, C_39_H_63_N_5_O_12_) was determined to be the same as that of **3** by positive-mode high-resolution electrospray ionization mass spectrometry (HRESIMS) analysis (*m*/*z* 794.4536 [M + H]^+^, calcd for C_39_H_64_N_5_O_12_ 794.4546, Δ − 1.3 ppm). The ^1^H NMR spectrum of **1** measured in CD_3_OD (Table 1) shared characteristic signals to that of **3**, i.e., two *O*-methyls at CH_3_-30 and -32 (*δ*_H_ 3.23, 3.56), two doublet methyls of CH_3_-27 and -28 (*δ*_H_ 1.18, 0.97), two singlet methyls at CH_3_-33 and -34 (*δ*_H_ 0.86, 1.00), an exomethylene unit at CH_2_-29 (*δ*_H_ 4.80, 4.64), acetal methylene bridging C-16 and C-18 (O-CH_2_-O, *δ*_H_ 4.79, 5.21) and a hemiacetal methine proton (N-CH-O, *δ*_H_ 5.80). Additionally, oxymethine protons at 11, 13, 15, 16, 17, 21 and 26 (*δ*_H_ 3.88, 4.24, 3.99, 4.17, 3.66, 3.47 and 3.65, respectively) were observed, suggesting that the common tricyclic core structure (C-13 to C-34) in **3** was also preserved in **1**. A spin system from H-2′ (*δ*_H_ 4.35) to H-5′ was suggested to be a part of arginine residue which was confirmed by the HMBC cross-peaks among H-2′/C-1 and H-5′/C-7′.

The distinct difference between **1** and **3** was observed in the side chain, as observed by the downfield-shifted signals for the polyene protons (H-2 to H-7). The overlapping signals of H-2 and H-3 were assigned based on the coupling constants of *J*_2,3_ (11.3 Hz), *J*_4,5_ (15.0 Hz) and *J*_6,7_ (15.0 Hz), which were obtained by processing data using a modified apodization function [16] (Figure 1). The geometry of the double bond between H-2 and H-3 in **1** was deduced as *cis*, confirming 2*Z*-onnamide A as **1**.

The molecular formula of 6*Z*-onnamide A (**2**, C_39_H_63_N_5_O_12_) was also determined to be the same as that of **3** by the positive-mode HRESIMS (*m*/*z* 794.4530 [M + H]^+^, calcd for C_39_H_64_N_5_O_12_ 794.4546, Δ − 2.0 ppm). The ^1^H NMR spectrum of **2** measured in CD_3_OD was superimposable with that of **3** (Table 2), except for the *cis* geometry in Δ^6,7^ (Figure 2), which indicated that **2** is 6*Z*-onnamide A.

The stereochemistry of 2*Z*- and 6*Z*-onnamides A (**1** and **2**) was assigned to the 11*R*, 13*R*, 15*S*, 16*R*, 17*S*, 18*S*, 21*S*, 22*R*, 25*R*, 26*R* and 2′*S* configuration, which is the same as onnamide A (**3**) and accounts for the biosynthetic pathway [8]. Analysis of the ^1^H NMR spectrum of **3** in CD_3_OD revealed that **3** undergoes photoisomerization by light irradiation to produce **1**, **2** and 4*Z*-onnamideA (**4**) (Appendix A). Calyculin and marinomycin analogs have been reported to undergo photoisomerization at their tetraene moieties [17,18]. Thus, we deduced that **1** and **2** are likewise artificially photoisomerized at the polyene moiety of **3**. However, it remains unclear whether the geometric isomerism occurs in living *T. conica* by sunlight or in the laboratory under artificial light. Metabolomic analysis using freshly collected *T. conica* should provide an answer to this issue.

Compounds **1**–**6** showed significant cytotoxicity against HeLa cells with IC_50_ values of 38–540 nM (Table 3). SARs study using HeLa cells revealed that compound **5** with the reduced C21–C22 single bond is as potent as onnamide A (**3**), whereas **6** with a shorter side chain showed weaker activity (×1/8.2). The positions of the *cis*-*trans* isomerism in the side chains of onnamides also affects cytotoxicity: compound **4** was 1.7-fold more potent than **3**, whereas **1** and **2** were 2.5-fold less cytotoxic than **3** (Figure 3).

Subsequently, the effects on histone modifications by onnamide A (**3**) were investigated using 16 monoclonal antibodies specific to each histone modification (Appendix A). The results revealed that **3** altered 13 histone modifications at concentrations of 70 and 140 nM, that are higher than the IC_50_ value (66 nM). Compound **3** enhanced the levels of trimethylated histone H3 lysine 4, 27 and 36 (H3K4me3, H3K27me3 and H3K36me3) and reduced the level of acetylated H4 lysine 5 (H4K5ac) at the lowest concentration (35 nM, Figure 4). All four histone modifications are related to cytotoxicity [19,20,21,22,23,24]. As H4K5ac is associated with DNA replication in the cell cycle, its decrease is consistent with the cell cycle arrest induced by **3** [14].

We also compared the effects of analogs **1**–**6** on histone modifications of H3K4me3, H3K27me3, H3K36me3 and H4K5ac at the same concentrations (35, 70 and 140 nM, Appendix A). Compound **6** administered at 70 nM (about 8 times less than the IC_50_ value of 540 nM) induced changes in the levels of the four histone modifications. However, enhanced cytotoxicity was also observed for **6** in this system (Appendix A), suggesting that the effects on histone modifications may be as the result of the cytotoxic effects by **6**.

Onnamide A (**3**) and anisomycin [25,26] were reported to inhibit protein synthesis and to elicit ribotoxic stress response (RSR) [27]. RSR is induced in response to ribosomal impairment in the mitogen-activated protein kinase (MAPK)-mediated inflammatory signaling cascade. It includes activation of stress-activated protein kinases (SAPKs), such as p38 and c-Jun *N*-terminal kinase (JNK), and eventually causes cell death [28,29]. Moreover, activation of SAPKs by anisomycin was reported to be independent of protein synthesis inhibition [25], implying the possible explanation for the difference in cytotoxicity by **3** and anisomycin.

Anisomycin and onnamide A (**3**) bind to different sites on the ribosome (anisomycin binds to the A site of the ribosome, whereas **3** binds to the E site) [30,31]. The different signaling pathways caused by different binding sites likely explain the weaker cytotoxicity of anisomycin [27]. We had expected that the effects on histone modification could reflect the difference in cytotoxicity, but the similar effects of anisomycin on histone modifications to those of **3** were observed (Appendix A).

In this study, we could confirm that both cytotoxicity and control of histone modifications change depend on the geometric isomerism in the side chains. Evaluating the effects on the control of histone modifications can be effective way to distinguish modes of action by two different types of cytotoxic compounds, although in this case, the largely overlapping mechanisms by anisomycin and onnamides in cytotoxicity hampered us from clearly distinguishing them.

## 3. Materials and Methods

### 3.1. General Experimental Procedures

NMR spectra were recorded on an Avance (400 MHz) spectrometer (Bruker Corporation, Billerica, MA, USA). ^1^H and ^13^C NMR chemical shifts were referenced to the solvent peaks, *δ*_H_ 3.31 and *δ*_C_ 49.15 for CD_3_OD (FUJIFILM Wako Pure Chemical Corporation, Osaka, Japan). HRESI-MS spectra were measured on a Triple TOF 4600 (AB Sciex Pte. Ltd., Tokyo, Japan) in the positive mode. Optical rotation was determined on a DIP-1000 digital polarimeter (JASCO Corporation, Tokyo, Japan) in CH_3_OH. UV spectrum was recorded using a UV-1800 spectrophotometer (Shimadzu Corporation, Kyoto, Japan). IR spectrum was measured on a JIR-WINSPEC50 spectrometer(JEOL Ltd., Tokyo, Japan). Fluorescent images were obtained with an IX70 microscope equipped by DP72 (Olympus Corporation, Tokyo, Japan).

### 3.2. Biological Material

The same *T. conica* specimens as previous work [32,33] were used in this study. *T. conica* was collected by hand using SCUBA, Amami-Oshima Is., Kagoshima Prefecture, Japan (N 28° 06.82′, E 129° 21.09′) in June 2007. The sample was immediately frozen and kept at −25 °C until extraction.

### 3.3. Isolation

The frozen sponge specimen (1020 g wet wt.) was extracted with CH_3_OH (1 L × 5), and the combined extract was evaporated in vacuo. The concentrated extract was suspended in H_2_O and extracted with CHCl_3_, then *n*-C_4_H_9_OH. The CHCl_3_ and *n*- C_4_H_9_OH layers were combined and subjected to the Kupchan procedure [15] yielding *n*-hexane, CHCl_3_ and aqueous CH_3_OH layers [32,33]. CHCl_3_ layer was concentrated to dryness and then separated by ODS flash chromatography (CH_3_OH/H_2_O = 5:5, 7:3, CH_3_CN/H_2_O = 7:3, 85:15, CH_3_OH, CHCl_3_/CH_3_OH/H_2_O = 6:4:1) to yield 6 fractions. The fraction eluting with CH_3_OH/H_2_O (7:3), which affected the histone modifications in HeLa cells, was separated by silica gel column chromatography (CHCl_3_, CH_3_Cl_3_/CH_3_OH = 19:1, 9:1, CHCl_3_/CH_3_OH/H_2_O = 8:2:0.1, 7:3:0.5, 6:4:1, 5:5:2). The fr.8 was further separated by reversed-phase HPLC (COSMOSIL 5C_18_-AR-II) using with CH_3_OH/H_2_O (6:4) to yield 12 fractions. Of these, 8.5 mg of onnamide B (**6**, tR: 24.6 min), 44.2 mg of onnamide A (**3**, tR: 37.0 min) and 4.3 mg of dihydroonnamide A (**5**, tR: 51.6 min) were isolated. The crude fr. 8 was separated by 2-step reversed-phase HPLC (COSMOSIL 5C_18_-AR-II, CH_3_OH/H_2_O = 32.5:67.5), followed by final purification with recycle system (phenomenex PHENYL-HEXYL, CH_3_CN/H_2_O = 4:6), to yield 3.5 mg of 2*Z*-onnamide (**1**, 11 cycles, tR: 155 min, 3.4 × 10^−3^% yield based on wet weights) and 1.5 mg of 6*Z*-onnamide A (**2**, 11 cycles, tR: 150 min, 1.5×10^−3^% yield based on wet weights). The crude Fr. 10 was purified by recycling reversed-phase HPLC (phenomenex PHENYL-HEXYL, CH_3_CN/H_2_O = 4:6) to afford 2.5 mg of 4*Z*-onnamides A (**4**, 6 cycles, tR: 87.0 min).

*2Z-onnamides A* (**1**): yellow amorphous solid; [α]_D_^22.8^ + 48.5° (*c* 0.2, CH_3_OH); UV (CH_3_OH) λ_max_ (logε) 299.4 (4.58) nm; IR (KBr film) ν_max_ 3363, 2936, 1652, 1635, 1576, 1558, 1531, 1397, 1093, 1032 cm^−1^; ESIMS *m*/*z* 794.4539 [M + H]^+^ (calcd for C_39_H_64_N_5_O_12_ 794.4546, Δ − 1.3 ppm); ^1^H and ^13^C NMR data, see Table 1.

*6Z-onnamides A* (**2**): yellow amorphous solid; [α]_D_^23.0^ +46.0° (*c* 0.2, CH_3_OH); UV (CH_3_OH) λ_max_ (logε) 300.4 (4.73) nm; IR (KBr film) ν_max_ 3417, 2925, 1644, 1583, 1537, 1403, 1316, 1092, 1032 cm^−1^; ESIMS *m*/*z* 794.4530 [M + H]+ (calcd for C_39_H_64_N_5_O_12_ 794.4546, Δ − 2.0 ppm); ^1^H and ^13^C NMR data, see Table 2.

*Onnamide A* (**3**): yellow amorphous solid; HRESIMS *m*/*z* 794.4542 [M + H]^+^ (calcd for C_39_H_64_N_5_O_12_ 794.4546, Δ − 0.5 ppm); ^1^H NMR data, see Appendix A.

*4Z-onnamides A* (**4**): yellow amorphous solid; HRESIMS *m*/*z* 794.4552 [M + H]^+^ (calcd for C_39_H_64_N_5_O_12_ 794.4546, Δ + 0.8 ppm); ^1^H NMR data, see Appendix A.

*Dihydroonnamide A* (**5**): yellow amorphous solid; HRESIMS *m*/*z* 796.4692 [M + H]^+^ (calcd for C_39_H_66_N_5_O_12_ 796.4702, Δ − 1.3 ppm); ^1^HNMR data, see Appendix A.

*Onnamide B* (**6**): yellow amorphous solid; HRESIMS *m*/*z* 768.4391 [M + H]^+^ (calcd for C_37_H_62_N_5_O_12_ 768.4389, Δ + 0.2 ppm); ^1^H NMR data, see Appendix A.

### 3.4. Cell Culture

HeLa human cervical cancer cells were cultured at 37 °C under an atmosphere of 5% CO_2_ in Dulbecco’s modified Eagle’s medium (DMEM, Low Glucose, FUJIFILM Wako Pure Chemical Corporation), containing 10% of fetal bovine serum (FBS, Biowest, Nuaillé, France), 2 µg/mL of gentamicin reagent solution AND 10 µg/mL of antibiotic-antimycotic. P388 murine leukemia cells were propagated and maintained at 37 °C under an atmosphere of 5% CO_2_ in Roswell Park Memorial Institute medium (RPMI, FUJIFILM Wako Pure Chemical Corporation), containing HRDS solution (2, 2′-dithiobisethanol), and kanamycin sulfate. The J1 mouse embryonic stem cells (ESCs) were obtained from the American Type Culture Collection (ATCC, Manassas, VA, USA) and maintained on 0.1% gelatin-coated dishes with mitomycin C-treated mouse embryonic fibroblasts (MEFs, Kitayama Labes, Nagano, Japan) in the medium of DMEM supplemented with 15% FBS, 1% L-glutamine (Gibco, Thermo Fisher Scientific, Inc., Waltham, MA, USA), 1% non-essential amino acids (Gibco), 1% penicillin-streptomycin (P/S, Gibco), 0.18% 2-mercaptoethanol (Gibco) and 1000 U/mL leukemia inhibitory factor (LIF, Merck Chemicals GmbH, Darmstadt, Germany).

### 3.5. Cytotoxic Test

Cytotoxicity test was conducted as previously reported [34]. Briefly, HeLa cells in DMEM or P388 cells in RPMI (cell concentration, 10,000 cells/mL, 200 µL) were added to each well of 96-well microplates and kept in the incubator at 37 °C under an atmosphere of 5% CO_2_. After 24 h, samples in DMSO with various concentrations of onnamides (**1**–**6**) were added to each well. After 72 h cultivation, to each well was added 50 µL of 3-(4,5-dimethyl-2-thiazoyl)-2,5-diphenyl-2H tetrazolium bromide (MTT) saline solution (1 mg/mL, FUJIFILM Wako Pure Chemical Corporation) and they were then kept in the incubator at 37 °C under an atmosphere of 5% CO_2_. After 4 h, medium was removed by aspiration and 150 µL of DMSO was added to each well to lyse cells. Concentration of the reduced MTT was quantified, measuring the absorbance at 650 nm to estimate IC_50_ values.

### 3.6. Histone Modification Assay

Assay of histone modification levels was performed by an immunofluorescence using a previously reported method [14], with some modifications. Briefly, HeLa cells were incubated under the medium containing the sample for 20 h and then immunostained. Cells were fixed with 4% paraformaldehyde in PBS for 10 min, 1% Triton X-100 in PBS for 20 min, and blocked in Blocking One-P (Nacalai Tesque Inc., Kyoto, Japan) for 20 min and then incubated in Alexa Fluor 488 (Thermo Fisher Scientific, Inc., Waltham, MA, USA) or Cy3 (Thermo Fisher Scientific, Inc.)-labeled antibodies against each histone modification (1:1000, Monoclonal Antibody Institute, Nagano, Japan) for 2 h with Hoechst 33342 (1:2000, Dojindo Laboratories Co., Ltd., Kumamoto, Japan). Fluorescent images were obtained with a microscope. The relative fluorescence intensity was digitalized using CellProfiler^TM^ software 3.0.0 [35] against those of the control wells. Anisomycin was purchased from FUJIFILM Wako Pure Chemical Corporation.

## 4. Conclusions

In conclusion, new onnamide analogs 2*Z*- and 6*Z*-onnamides A (**1** and **2**) and four known onnamides (**3**–**6**) were isolated from the marine sponge *T. conica*, and their structures were elucidated by MS and NMR spectral analyses. The combined effects on histone modifications and cytotoxicities by these compounds revealed that their modes of action follow those of anisomycin. Identifying modulations in the expression patterns of target genes caused by onnamides via changes in histone modifications should provide new insights into the underlying mechanisms of onnamide activity. A detailed investigation of these mechanisms is currently underway.

## Figures and Tables

**Figure 1 molecules-28-02524-f001:**
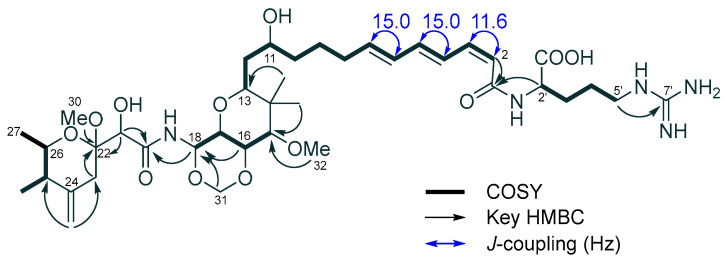
COSY, key HMBC cross-peaks and coupling constants of 2*Z*-onnamide A (**1**).

**Figure 2 molecules-28-02524-f002:**
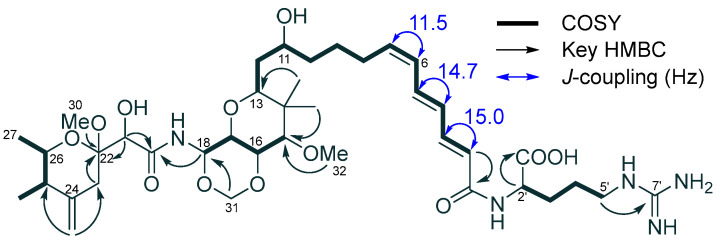
COSY, key HMBC cross-peaks and coupling constants of 6*Z*-onnamide A (**2**).

**Figure 3 molecules-28-02524-f003:**
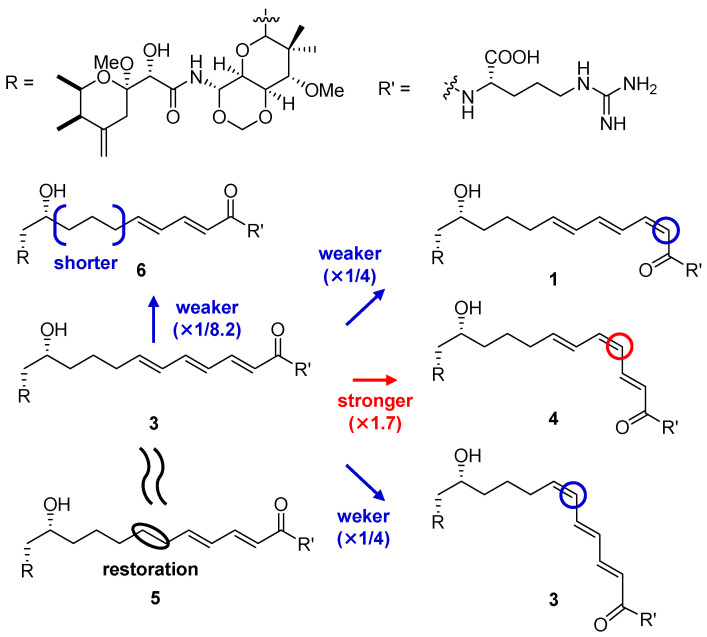
SARs of compounds **1**–**6** on the cytotoxicity against HeLa cells.

**Figure 4 molecules-28-02524-f004:**
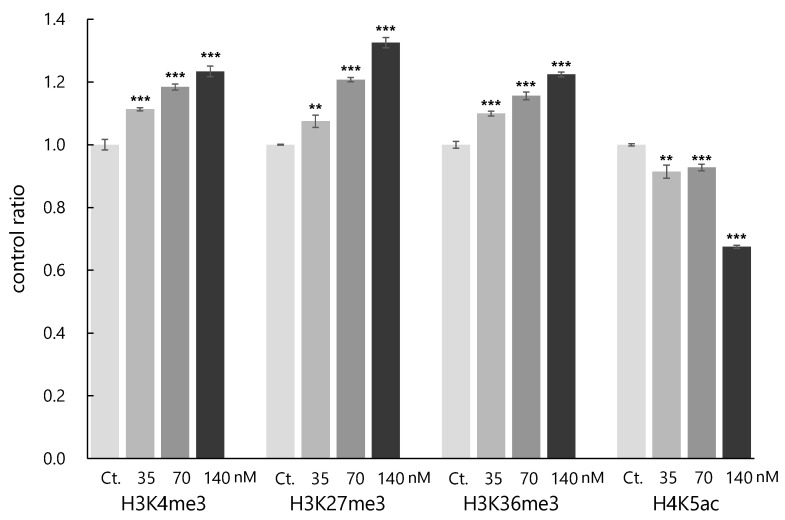
Effects of onnamide A (**3**) on histone modifications. Quantification of the histone modification levels of H3K4me3, H3K27me3, H3K36me3 and H4K5ac after cultivation in medium containing samples for 20 h (samples: 35, 70 and 140 nM of **1**, Ct.: DMSO, *n* = 3, mean ± S.D. ***: *p* < 0.001, **: *p* < 0.01, Dunnett test).

**Table 1 molecules-28-02524-t001:** NMR spectral data for 2*Z*-onnamide (**1**) in CD_3_OD (400/150 MHz).

Position	*δ_C_*	*δ_H_* Mult. (*J* in Hz)	COSY	HMBC
1	168.5			
2	120.8	5.74 d (11.6)	H-3	C-1, C-4
3	141.7	6.46 t (11.6)	H-2, H-4	C-1, C-5
4	128.4	7.45 dd (15.0, 11.6)	H-3, H-5	
5	142.4	6.43 dd (15.0, 10.7)	H-4, H-6	
6	131.9	6.24 dd (15.0, 10.7)	H-5, H-7	
7	140.5	5.95 dt (15.0, 6.9)	H-6, H-8	C-5
8	34.1	2.13 m, 2.23 m	H-7, H-9	
9	31.6	1.30 m, 1.47 m	H-8, H-10	
10	37.0	1.29 m,1.43 m	H-11, H-9	
11	71.2	3.65 m	H-10, H-12	
12	37.5	1.54 m	H-11, H-13	
13	78.8	3.47 dd (8.9, 3.4)	H-12	
14	42.5			
15	80.7	3.66 d (9.9)	H-16	C-32
16	75.8	4.17 dd (9.9, 6.6)	H-15, H-17	C-15, C-17, C-18, C-31
17	71.0	3.99 dd (9.4, 6.6)	H-16	
18	75.0	5.80 d (9.4)	H-17	C-20, C-31
20	174.6			
21	74.1	4.24 s		C-20, C-23, C-24
22	101.5			
23	34.9	2.41 d (14.3), 2.32 d (14.3)		C-22, C-24, C-25, C-29
24	148.4			
25	43.2	2.20 m	H-26, H-28	C-24
26	71.0	3.88 qd (6.5, 2.5)	H-25, H-27	C-25
27	18.3	1.18 d (6.5)	H-26, H-28	C-25,26
28	12.5	0.97 d (7.0)	H-25, H-27	C-24, C-25, C-26
29	110.2	4.80 brs, 4.64 brs		C-23, 25
30	48.8	3.23 s		C-22
31	87.8	5.21d (6.8), 4.79 d (6.8)		C-18
32	62.1	3.56 s		C-15
33	14.3	0.86 s		C-13, C-14, C-15, C-34
34	23.7	1.00 s		C-13, C-14, C-15, C-33
1′	178.6			
2′	55.2	4.35 dd (7.3, 5.2)	H-3′	C-1, C-1′, C-3′
3′	31.6	1.89 m, 1.74 m	H-2′, H-4′	C-1′
4′	26.3	1.65 m	H-3′, H-5′	C-3′, C-5′
5′	42.3	3.19 m, 3.26 m	H-4′	C-7′
7′	158.7			

**Table 2 molecules-28-02524-t002:** NMR spectral data for 6*Z*-onnamide (**2**) in CD_3_OD (400/150 MHz).

Position	*δ_C_*	*δ_H_* Mult. (*J* in Hz)	COSY	HMBC
1	168.5			
2	120.8	6.11 d (15.0)	H-3	C-1, C-4
3	141.7	7.25 dd (15.0, 11.2)	H-2, H-4	C-5
4	128.4	6.35 dd (14.7, 11.2)	H-3, H-5	
5	142.4	6.94 dd (14.7, 11.5)	H-4, H-6	
6	131.9	6.13 t (11.5)	H-5, H-7	
7	140.5	5.67 dt (11.5, 7.7)	H-6, H-8	
8	34.1	2.30 m	H-7, H-9	
9	31.6	1.50 m, 1.58 m	H-8, H-10	
10	37.0	1.32 m	H-11, H-9	
11	71.2	3.65 m	H-10, H-12	
12	37.5	1.54 m	H-11, H-13	C-17
13	78.8	3.49 m	H-12	
14	42.5			
15	80.7	3.66 d (9.8)	H-16	C-32
16	75.8	4.17 dd (9.8, 6.6)	H-15, H-17	C-15, C-17, C-18, C-31
17	71.0	3.98 dd (9.1, 6.6)	H-16	
18	75.0	5.83 d (9.1)	H-17	C-20
20	174.6			
21	74.1	4.24 s		C-20, 23, 24
22	101.5			
23	34.9	2.41 d (14.3), 2.32 d (14.3)		C-22, 24, 25, 29
24	148.4			
25	43.2	2.20 m	H-26, H-28	
26	71.0	3.87 qd (6.5, 2.6)	H-25, H-27	
27	18.3	1.17 d (6.5)	H-26	C-25,26
28	12.5	0.97 d (7.1)	H-25	C-24, 25, 26
29	110.2	4.79 s, 4.64 brs		C-23, 25
30	48.8	3.24 s		C-22
31	87.8	5.23d (6.9), 4.80 d (6.9)		C-18
32	62.1	3.56 s		C-15
33	14.3	0.86 s		C-13, 14, 15, 34
34	23.7	1.00 s		C-13, 14, 15, 33
1′	178.6			
2′	55.2	4.38 dd (7.3, 5.2)	H-3′	C-1′
3′	31.6	1.89 m, 1.75 m	H-2′, H-4′	
4′	26.3	1.64 m	H-3′, H-5′	C-2′, 3′
5′	42.3	3.18 m, 3.22 m	H-4′	C-7′
7′	158.7			

**Table 3 molecules-28-02524-t003:** Cytotoxicity against HeLa and P388 cells by compounds **1**−**6**.

Cell Line	IC_50_ (µM)					
	1	2	3	4	5	6
HeLa	0.17	0.15	0.066	0.038	0.057	0.54
P388	1.8	4.8	0.62	0.31	0.57	5.2

## Data Availability

Data from the present study are available in the article and Appendix A.

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
