# Peer review of "Two Onnamide Analogs from the Marine Sponge Theonella conica: Evaluation of Geometric Effects in the Polyene Systems on Biological Activity"

_molecules, 2023, doi:10.3390/molecules28062524_

Round 1

Reviewer 1 Report

The authors present a manuscript entitled “Two onnamide analogs from the marine sponge Theonella conica: evaluation of geometric effects in the polyene systems on biological activity.” The two new molecules have a different geometry of double bond in 2 and 6 position with the Z geometry. The structures were elucidated using NMR and HRMS. Moreover, analysis of coupling constants showed clear evidence. The author has a speculation that the two new molecules are artifact of E geometry (compound 3) by photoisomerization mechanism and give an example of calyculin cases. In my opinion, it is possible to have phototoisomerization reaction as in the case on linear molecule (calyculin analogs) or macrocyclic rings (marinomycins). The more the molecule has conjugated double bonds, the more reactive the molecules. It has been showed by synthetic studies that marinomycins was able to have a photoisomerization reaction in order of time 0-120 minutes. The SAR of the tested molecules showed that the geometry in the side chain of onnamides also affects cytotoxicity. In this case, although the biosynthetic origin of the Z geometry molecules has not yet revealed, the author expected that the molecules are naturally occurring compounds. The plus is that the two molecules can give more understanding of the geometry effect of the onnamide side chains on their cytotoxicity. I recommend to accept this manuscript after grammatical check. 

Reviewer 2 Report

The cell line test and toxicity results should be presented within the main manuscript 

Reviewer 3 Report

The researchers studied the composition of the marine sponge Theonella conica, resulting in the discovery of two new onnamide analogs named 2Z- and 6Z-onnamides A (1 and 2) together with four known analogs 36. Their structures were characterized by the geometries of their triene moieties. Moreover, all of the isolates exhibited different levels of histone modifications and cytotoxicity against HeLa and P388 cells, indicating the geometric effects on biological activities. Based on these intriguing findings, this manuscript will gain interests from readers. However, revisions are necessary before this manuscript is considered to be accepted.

Comments:

1. For Introduction, extensive modifications are required. Marine invertebrates are composed of many phyla including the Porifera, and Theonella is a genus of Porifera. What is the structure type of onnamides? Were they peptides, polyketides or terpenoids? Were HeLa or P388 cells used in the previously reported potent cytotoxicity bioassays? Were any secondary metabolites of Theonella sponges screened for histone modifications? After reading the introduction, these questions come to mind. It is better to clearly present in the revised version. And it is more appropriate to include the last paragraph in Conclusions.

2. For the structure elucidation of 1, the spectral date of the arginine-like terminal of side chain should be analyzed. Then, on the basis of characteristic signals of all fragments, the similarity and difference between compounds 1 and 3 could be found.

3. Only the relative configurations of these two new compounds were reported. For readers, they are more interested in their absolute configurations. Have you tried? Was the absolute configuration of any onnamides reported ambiguously? Since the biosynthesis of onnamide A had been proposed in Ref. [14], it is likely the absolute configuration of onnamide A was verified.

Others:

1. P2L58: As shown in Figure S1, the observed m/z was 794.4539 not 794.4536.

2. Figure 1: The numbering 2 should be put on the carbon adjacent to the carboxyl.

3. Tables 1 & 2: Please keep all the 13C NMR data with one digit after the decimal point.

4. Figure 2: The numbering 6 was labeled wrongly.

5. P5L98: The results of cytotoxicity bioassay were recorded in Table S3 not Table S2.

6. Supporting Information: The captions of Tables S2 and S3 should be exchanged in the legend on the first page.

7. Isolation: Please add tR values for those compounds isolated by HPLC.

Reviewer 4 Report

Most of the references are old. There is even one reference from 1974. 

Not so many references from the last 3 years.

I suggest to update the reference list.
